# Could BMPs Therapy Be Improved if BMPs Were Used in Composition Acting during Bone Formation in Endochondral Ossification?

**DOI:** 10.3390/ijms231810327

**Published:** 2022-09-07

**Authors:** Anna Hyc, Anna Osiecka-Iwan, Stanislaw Moskalewski

**Affiliations:** Department of Histology and Embryology, Medical University of Warsaw, 02004 Warsaw, Poland

**Keywords:** BMP therapy, endochondral bone formation, growth factors, human research, ethical standards

## Abstract

The discovery of bone morphogenetic proteins (BMPs) inspired hope for the successful treatment of bone disorders, but side effects worsening the clinical effects were eventually observed. BMPs exert a synergistic effect, stimulating osteogenesis; however, predicting the best composition of growth factors for use in humans is difficult. Chondrocytes present within the growth plate produce growth factors stored in calcified cartilage adhering to metaphysis. These factors stimulate initial bone formation in metaphysis. We have previously determined the growth factors present in bovine calcified cartilage and produced by rat epiphyseal chondrocytes. The results suggest that growth factors stimulating physiological ossification are species dependent. The collection of human calcified cartilage for growth factors determination does not appear feasible, but chondrocytes for mRNA determination could be obtained. Their collection from young recipients, in view of the Academy of Medical Royal Colleges Recommendation, would be ethical. The authors of this review do not have facilities to conduct such a study and can only appeal to competent institutions to undertake the task. The results could help to formulate a better recipe for the stimulation of bone formation and improve clinical results.

## 1. Introduction

The isolation of bone morphogenetic proteins from bovine material and the production of their human equivalents by genetic engineering inspired hope that their application would considerably improve the treatment of bone disorders [1]. BMPs and other growth factors act on mesenchymal stem cells (MSCs), stimulating their differentiation into osteoprogenitor cells, as well as the differentiation of the latter into osteoblasts and osteoclasts [2,3,4,5,6,7,8,9,10,11,12,13,14]; thus, they were obvious candidates for clinical use. Unfortunately, their application in numerous instances was disappointing and caused various harmful side effects [15]. Our previous studies suggested that the panel of growth factors acting during the initial stages of physiological bone formation is species specific [16,17]. Clinically used BMP-2 and 7, as well as several other growth factors, effectively produce bone in rats [1] but selecting them for use in clinics was arbitrary since there were no data to decide which would be optimal for humans. During endochondral bone formation, growth factors are produced by chondrocytes, deposited in a cartilage calcified and non-calcified matrix, released, and used for the stimulation of osteogenesis. Determining which growth factors are produced by human chondrocytes and dominate during endochondral ossification could at least offer a hint regarding which of them would be best for healing bone disorders. In this review, we outline the processes occurring during the initial stages of bone formation in the hope that it will encourage studies on human chondrocytes obtained from young transplant donors. Describing the mechanism of BMPs action and the role of BMPs in osteogenesis and osteoblast differentiation, we tried to further underline the need for recognizing factors specific for bone formation in humans. The use of such factors—if they were recognized—could improve the treatment of bone disorders; however, at present, as evaluated by specialists and described in the paragraph “BMPs in clinics—pros and cons”, they are far from optimal.

The detection of growth factors responsible for physiological bone formation in humans could be realized in the framework of translational medicine. This procedure consists of three components: the scientists studying particular problems, the clinicians introducing the results of basic studies into medical practice, and community supplying funds for the realization of the program [18,19]. In this particular case, since the material for growth factors detection would have to originate from human donors, the willingness of a donor family to donate tissues for medical studies is paramount. 

## 2. Mechanism of Bone Morphogenetic Proteins (BMPs) Action

The transforming growth factor beta (TGFβ) superfamily contains more than 30 members, including TGFβs, bone morphogenetic proteins (BMPs), growth differentiation factors (GDFs), and activins, that work through a wide variety of signalling pathways, i.e., through different ligands, receptors and signalling molecules [7,20,21,22,23,24]. The name BMP refers to molecules that evoke the activation of the canonical BMP signalling pathway, restricting the list to approximately 15 BMP ligands in humans [9,23]. Ligands from the BMP family can form homodimeric or heterodimeric complexes, except BMP3 and BMP15, which work as monomers. Moreover, the ability to induce bone formation and the signalling action of homodimers BMP2, BMP4, BMP5, BMP6, and BMP7 increases up to 50 times when they act in heteromeric forms, such as BMP2/5, BMP2/6, BMP2/7, and BMP4/7. This mechanism is associated with the higher affinity of BMP receptors to BMP heterodimers [9]. Bone morphogenetic protein receptors (BMPRs) are transmembrane, receptors with the activity of serine-threonine kinase divided into type I and II. ACVR1 (activin A receptor, type I or ALK2 activin receptor-like kinase-2), ACVR1B (ALK-4), ACVR1C (ALK-7), ACVRL1 (ALK1), BMPR1A (bone morphogenetic protein receptor, type IA or ALK3), BMPR1B (bone morphogenetic protein receptor type-IB or ALK6) and TGFβR1 (TGF beta receptor I or ALK5) belong to type I. Type II BMP receptors include BMPR2 (bone morphogenetic protein receptor type II), ACVR2A (activin receptor type-2A), ACVR2B (activin receptor type-2B), TGFBR2 (TGFβ receptor II), and AMHR2 (anti-Mullerian hormone receptor type 2) [10]. BMP dimers attach to BMP receptors with variable degrees of affinity. BMP2 and BMP4 preferentially bind ALK3 and 6 in a complex with BMPR2, ACVR2A or ACVR2B. BMP6 and 7 mainly attach ALK2 and ACVR2A, while BMP9 and 10 mainly attach ALK1 in a complex with BMPR2 or ACVR2B [20,24]. Signal transduction into the cell requires the attachment of BMP ligand to type I and type II receptors. Mature receptor signalling complex is composed of ligand dimer and two type I and two type II receptors (tetramer). Constitutively active type II receptors are responsible for the phosphorylation of type I receptors to induce canonical and non-canonical BMP signalling pathways [20,24]. In the canonical pathway, activated type I receptors phosphorylate and activate Smad1, 5, and 8 (R-Smad) recruit co-Smad (Smad4) to form heteromeric complexes. Then, such complexes are transported into the nucleus and recruit transcription factors, e.g., Hox-C8 (homeobox protein C8), and RUNX2 (runt-related transcription factor 2) and regulate the expression of significant osteogenic genes [9,10]. Smad proteins (homologues of the Drosophila protein, mothers against decapentaplegic (Mad) and the Caenorhabditis elegans protein Sma) include R-Smads (receptor-regulated Smads), i.e., Smad1, Smad5 and Smad8 [7] and Smad4 (Co-Smad common partner Smad). The complex of Smad1/5/8 with Smad 4 is translocated to the nucleus. Smad4 contains two highly conserved MH1 and MH2 domains (Mad homology 1 and 2) that play a critical role in DNA recognition/binding as well as in BMP receptor interaction. The complex of Smads binds to Runx2, an essential transcription factor in osteogenesis, and controls osteoblastic gene expression and differentiation [5] (Figure 1).

In the non-canonical pathway, activated type I receptors phosphorylate MAP3K7 (mitogen-activated protein kinase kinase kinase 7 or TAK1—TGFβ-activated kinase 1) recruit MAP3K7IP1 (MAP3K-interacting protein 1 or TAB1—TAK1 binding protein 1) and initiates a p38 MAPK signalling cascade. The phosphorylated form of p38 MAPK phosphorylates and activates RUNX2, DLX-5 (distal-less homeobox 5), and Sp7 or Osx (specificity protein 7 or osterix) transcription factors to evoke the expression of osteogenic genes [9,10]. For example, activation of the MAPK-Smad2/3 pathway by TGFβ can stimulate osteoprogenitor proliferation and osteoblastic lineage formation. Activation of MAPK by BMPs can induce differentiation of osteoprogenitors into osteoblasts [7,21] (Figure 1).

## 3. BMPs in Osteogenesis

The activity of BPMs, triggered in proper order, is crucial in both endochondral and intramembranous bone formation. Ossification begins from the condensation and differentiation of mesenchymal stem cells (MSCs). Studies in mouse models and cell cultures showed that numerous BMPs, such as BMPs 2, 4, 5, and 7, are responsible for the differentiation of MSCs into chondrocytes. BMPs acting through Sox9 (SRY-box transcription factor 9) cause the production of genes characteristic for chondrocytes, e.g., type II collagen [9,25]. BMP7 occurs in proliferating chondrocytes, particularly close to the perichondrium. Later BMPs 2 and 6 secreted by hypertrophic chondrocytes activate RUNX2. BMPs induce osteoblast differentiation directly from MSCs. BMPs 2, 6, and 9 are the key regulators of the osteoblast lineage-specific differentiation from MSCs [9]. BMP2 activates the expression of RUNX2 and Sp7, obligatory transcription factors for osteoblast differentiation, and induces osteoblastic markers such as alkaline phosphatase (ALP), type I collagen, and osteocalcin (OCN). BMP-7 stimulates the mineralization of the extracellular matrix and activity of ALP activity [26].

BMP2 stimulates osteoclastogenesis, causing the differentiation of osteoclast precursors into osteoclasts. BMP2 mainly binds to the type I receptors, particularly BMPR1A [8]. The action of activated BMP2 is under the control of regulatory feedback mechanisms. One of them acts through BMP antagonists like Noggin. Noggin inhibits BMP signalling by attaching to critical epitopes on both the type I and type II receptors and blocking their activity [8,27,28].

Since BMPs bind to BMP receptors with varying degrees of affinity, it is evident that the optimal stimulation of osteogenesis in clinical practice would occur if the BMP stimulant matched the receptors of the target cells. To achieve this goal, growth factors dominating in human bone formation under physiological conditions must be recognized.

## 4. Osteoblast Differentiation

As the starting material for studies on osteoblast differentiation often serve bone marrow-derived mesenchymal stem cells (BMMSCs), which, when placed in culture, can differentiate into osteoblasts, chondrocytes and adipocytes, the International Society for Cellular Therapy (ISCT) suggested basing the identification of MSCs on the identification of surface markers such as CD36, CD44, CD73, CD90 and CD105. In confluent, long lasting cultures of MSCs in osteogenic medium mineral deposition was observed, indicating the early stages of bone formation [13]. MSCs may also be stimulated toward osteogenic differentiation in culture by ascorbic acid and dexamethasone, or by numerous growth factors, mainly belonging to the BMPs and wingless-related integration site (WNT) pathways [11,13].

Fu et al. [29] prepared a list of other factors stimulating the mobilization of BMMSCs from bone marrow into circulation and their migration into the site of damage, i.e., stromal derived factor-1 (SDF-1), osteopontin (OPN), basic fibroblast growth factor (bFGF), vascular endothelial growth factor A (VEGFA), hepatocyte growth factor (HGF), insulin-like growth factor 1 (IGF1), platelet-derived growth factor (PDGF) and TGFβ1. It is particularly interesting that osteopontin augmented the expression of integrin β1 in BMMSCs and stimulated their migration through the binding to integrin β1. Moreover, it could induce the rearrangement of actin cytoskeleton, resulting in the reduced stiffness of cytoplasm. OPN also reduced the stiffness of the nucleus, decreasing the production of lamin A/C, the main factors of nuclear stiffness. Greater flexibility of the nucleus facilitates the migration of MSCs through narrow openings in capillary walls. All of the above-mentioned growth factors are necessary for inducing MSCs homing to the site of the injured area and for the engagement of MSCs in the repair of damaged tissue [29].

Mesenchymal cells differentiation into osteogenic precursors and osteoblasts requires the stimulation of various molecular factors, particularly those acting through the BMPs and WNT pathways [11,30]. In the first step, osteogenic transcriptional factors in MSCs are activated. They include RUNX2 [31], Sp7 [32] and DLX5 [33]. One of the pathways activated by BMPs is connected with Indian hedgehog homolog (IHH) signalling activation, which also stimulates osteogenic transcriptional factors [34]. RUNX2 directs mesenchymal stem cells to differentiate into osteoblast progenitors, while RUNX2, Sp7 and DLX5 direct osteoblast progenitors to differentiate to immature osteoblasts. These pre-osteoblasts express osteogenic genes encoding for alkaline phosphatase, collagen1α1 chain (COL1A1), osteopontin, bone sialoprotein II (BSP II) and osteocalcin and differentiate into osteoblasts and osteocytes [11]. RUNX2 and Sp7 are recognized as essential for the differentiation of osteoblast; their deletion leads to a lack of osteoblasts during both intramembranous and endochondral bone formation [32]. RUNX2 and Sp7 are responsible for the upregulation of the transcription of numerous osteoblast-related genes. RUNX2 causes the expression of COL1A1, OPN, OCN and IBSP (integrin binding sialoprotein) [4]. Sp7 upregulates the expression of collagens (COL1A1, COL5A1, COL5A3), VEGF, fibromodulin (Fmod), matrix metalloproteinase 9 and 13 (MMP9 and 13); pyrophosphatase/phosphodiesterase 1 (ENPP1), integrin binding sialoprotein (IBSP), sclerostin, osteocalcin, connexin 43 (Cx43) or zinc transporter 1 (ZIP1) [32]. Acetylation of Sp7 increases its binding to the promoters of genes for ALP, BSP, COL1A1 and osteocalcin, demonstrating that Sp7 acetylation is important for the differentiation of osteoblasts [33].

There still remains one attribute of the osteogenic cells not even approached in molecular studies: their memory of the site of origin. Osteoblasts isolated from calvarial or endochondral bone produce in intramuscular transplants bone, similar to that in the organs of their origin [35].

A new area of osteoblast differentiation paved the way for observations on the involvement of miRNAs. Several miRNAs target the WNT and BMP pathways, thus inhibiting or stimulating the differentiation of osteoblasts [11]. Interplay between the microRNAs and WNT, TGFβ, and BMP signalling pathways stimulate mesnchymal stem cells to differentiate to the osteoblasts [36,37]. miRNAs that target RUNX2 corepressors (Snail 1—zinc finger protein, HDEAC—histone deacetylases 3,4,5,6) favour osteogenesis, while miRNAs that target Runx2 coactivators (DLX5) inhibit bone formation [38]. The miRNAs may also directly target the TGFβ superfamily ligands and regulate osteoblastic differentiation by targeting TGFβR1 and TGFβR2, as well as BMPR1 and BMPR2 signalling receptors [30].

Moreover, miRNA participates in the commitment of bone marrow precursors toward osteoclasts and plays a role in the early and late phase of osteoclastogenesis [39,40]. More than 20 miRNAs are involved in various stages of osteoclast formation, e.g., miR-124 regulates RANKL-dependent and -independent differentiation of osteoclasts, while miR-218 exerts a contrary effect [41].

The BMP signalling pathway controls diverse cellular activities, modulating the levels of miRNAs. BMP2 downregulating miRNA141 and miRNA200a expression stimulate pre-osteoblast differentiation. On the other hand, miRNA200a is a positive regulator activating BMP signalling during the differentiation of bone cells [42]. The question arises whether miRNAs of osteoprogenitor cells from various species respond in the same fashion to particular growth factors or whether the response is species specific. If so, it would be particularly important to establish which growth factors dominate physiological bone formation in humans.

Another mechanism affecting osteoblast differentiation involves the secretion of exosomes (extracellular vesicles, EVs) defined as the diverse, nanoscale membrane vesicles actively released by cells [43]. Some exosomes are involved in cartilage or bone mineralization [44]. Exosomes produced by bone marrow stem cells (BMMSCs) may stimulate osteoblast differentiation by upregulating RUNX2. Its upgrade leads to the production of BMMSC-EVs able to stimulate the healing of damaged bone [11,45,46]. EVs may also serve as the transporters of miRNA from osteoclasts to mesenchymal stem cells, osteoblasts, osteocytes and chondrocytes [41].

The formation of osteoblasts is also regulated by a network of cytokines that may serve as potential action sites for the treatment of bone related diseases. Interleukin 10 (IL10), IL11, IL18, or interferon γ (IFNγ) belong to cytokines stimulating osteoblast formation, while tumour necrosis factor-α (TNFα), TNFβ, IL1α, IL4, IL7, IL12, IL13, IL23, IFNα, IFNβ, and many others downregulate osteoblastogenesis. Proinflammatory cytokines, e.g., TNF and IL1α, inhibit osteoblastogenesis, mainly by the downregulation of RUNX2 or Sp7 expression. Osteoblastogenic cytokines, e.g., IL18, IFNγ, induce RUNX2-mediated osteoblasts differentiation and proliferation [2]. Some of the cytokines mentioned above, e.g., TNFα and IL10, have been demonstrated to act through the regulation of the activation of proteins belonging to the BMP family [47].

## 5. BMPs in Clinics—Pros and Cons

The hope for a marked improvement in bone healing with the use of endogenous growth factors was raised by the pioneer work of Urist [48], who described the presence of bone morphogenetic proteins in decalcified bovine bone. This finding resulted, after years of studies, in their isolation [1,12,49]. At present, more than twenty BMPs have been identified, from which at least eight are able to evoke cartilage and/or bone formation after intramuscular transplantation or in in vitro tests [50]. Additionally, GDF5, another member of the BMPs family, has distinct osteoinductive properties [51]. Moreover, another significant factor NELL1 (NEL-like protein 1) participates in bone repair and regeneration [52].

Two BMP products, human recombinant BMP2 and BMP7, are allowed to be used in clinics in the USA [12]. Early studies on the clinical application of bone morphogenetic proteins suggested that they may provide for revolutionary therapies in orthopaedic practice [1]. In the reviews published in 2005 [53] and 2007 [54], authors concluded that the bone morphogenetic proteins may be safely used in treatment of bone disorders. In the review from 2016 [15], however, James at al. summarized the side effects of BMP2 treatment, including postoperative inflammation, ectopic bone formation, osteoclast-mediated bone resorption, and inappropriate adipogenesis, as well as the life-threatening spine swelling in cervical surgery. In a rat experimental study, high doses of BMP2, equivalent to the doses used in clinical practice (1500 µg/mL), consistently caused cyst-like bone formation and soft tissue swelling [55]. These side effects could be dependent on the quality of the formulation and administration methods, or on the instability of BMPs. After better control of these factors is achieved, considerable improvements may be expected [56]. On the other hand, BMP2 is usually upregulated in various, tumours and is associated with tumour progression, [57,58]. Moreover, various BMPs and TGFβs, stimulate angiogenesis and may be involved in the formation of multiple malignancies, including breast cancer, lung, adrenal, and colorectal tumours, which raises potential concerns regarding their clinical use [59].

## 6. Synergistic Use of Various BMPs

The idea that the synergistic use of various BMPs might improve effective bone regeneration has already been expressed by various authors. Celeste et al. [60] suggested the additive or synergistic contribution of BMP5, BMP6, and BMP7 to BMP2 for the de novo induction of in vivo bone formation. Wutzl et al. [61] found that in cultures of mouse bone marrow cells, BMP2 alone and in combination with BMP2 and BMP5 significantly enhanced osteoclastogenesis, while BMP2, BMP5, and BMP6 used jointly did not exert additional effects. However, BMP2, BMP5, and BMP6 used together positively influenced, matrix mineralization and Sp7 expression, a transcriptional factor characteristic for mesenchymal cells which differentiate into osteoblasts. Cheng et al., [6] on the basis of studies with mouse pluripotential mesenchymal precursor and preosteoblastic cell lines, suggested a hierarchical model with BMP2, 6, and 9, inducing the differentiation of mesenchymal stem cells into osteoblast, while BMP2, 4, 6, 7, and 9 stimulate osteogenesis. Luu et al. [62] showed that besides BMP2 and BMP7, BMP6, and BMP9 from all BMPs have the highest osteogenic activity, both in vitro in mouse cell lines and in vivo in mouse intramuscular transplants. The authors suggested that osteogenic BMPs (i.e., BMP2, 4, 6, 7, and 9) may be used to formulate synergistic pairs among themselves and/or with other less osteogenic BMPs for efficacious bone regeneration in humans. Friedman et al. [63] reported that for human mesenchymal stem cells (hMSC) cultured in defined, serum-free conditions and treated with BMP2, 4, 6, and 7, BMP6 was the most consistent and potent regulator of osteoblast differentiation. When BMPs were used jointly, only combinations containing BMP6 promoted strong mineralization. BMP6 was also superior to BMP2 and BMP7 in stimulating the differentiation of osteoblasts in vitro and bone formation in vivo [64].

While all these studies stress the benefits of the joint use of various BMPs for bone formation, the suggested recipes are formulated on the basis of experimental data from animal systems. Therefore, it is difficult to assess whether they would be optimal for human beings. Considering the large number of possible combinations, it would be extremely difficult to evaluate them in practice. According to Vukicevic and Grgurevic [65], BMP6 is present in human blood and is produced by bone marrow stem cells before their differentiation into osteoblasts. Moreover, it is also secreted by osteoclasts as a chief bone coupling factor, attracting osteoblasts to the resorption site. Thus, it is suggested that BMP6 participates in physiological ossification in humans; however, even if this is true, the question remains regarding whether it acts alone, or is supported by other BMPs or NELL1.

## 7. Endochondral Ossification in Fracture Healing

Thus, going back to the title question—could BMPs therapy be improved by imitation of bone formation in endochondral ossification? The affirmative answer requires the assumption that the reaction of osteoprogenitor cells from the epiphyseal growth plate to growth factors is similar to that of osteoprogenitors present in periosteum or bone marrow in a mature organism. This assumption seems supported by observations of fracture healing, usually involving the formation of mineralizing cartilaginous callus, which later is resorbed and replaced with bone, i.e., processes with a strong resemblance to those occurring within the growth plate [66]. It was also demonstrated that various pro-osteogenic factors such as PDGF, TGFβ, FGF1, IGF, and BMPs, and pro-angiogenic factors such as VEGF, BMPs, FGF and TGFβ participate in the regeneration of damaged bone [67]. Some authors advise direct delivery of MSCs from bone marrow to the regenerating sites of the damaged bone, expecting that they will differentiate along chondrogenic and/or osteogenic lines under the influence of local growth factors and improve clinical results [22].

## 8. How Epiphyseal Growth Plate Produces and Stores Growth Factors

Cartilage within the epiphyseal growth plate is divided into a reserve zone, with stem cells differentiating and rapidly dividing in the proliferative zone, enlarging in the hypertrophic zone, and finally undergoing apoptosis in the provisional calcification zone close to the metaphysis. The rows of chondrocytes in the proliferative and hypertrophic zone are separated by a cartilage matrix forming calcified longitudinal septa, while groups of chondrocytes within rows are partitioned by non-calcified transverse septa [68] (Figure 2).

Calcification of longitudinal septa begins in the middle part of the proliferative zone due to the activity of matrix vesicles; detached from chondrocytes and carrying alkaline phosphatase, components necessary for calcium phosphate deposition and also BMPs 1–7 and VEGF [69]. In the provisional calcification zone, about 70% of the matrix is calcified, with the remaining matrix either adhering to or enclosed within calcium deposits [70,71].

## 9. Osteoclasts Release Growth Factors from Calcified Cartilage and Transport Them towards Osteoprogenitor Cells

Growth factors stored in the zone of provisional calcification (in calcified and non-calcified cartilage) may serve as stimulants of bone formation in metaphysis. Calcified septa are partially resorbed by cells migrating from the metaphysis and differentiating into multinuclear cells called osteoclasts or chondroclasts [71,72]. The non-resorbed part of the calcified longitudinal septa serves as the place for initial bone deposition. The osteoclasts are followed by endothelial cells forming new capillaries. Mesenchymal cells follow the vascular elements, differentiate into osteoblasts and deposit osteoid on the remnants of the non-resorbed longitudinal septa [71,72,73,74].

The resorption of calcified cartilage by osteoclasts involves the rearrangement of their cytoskeleton. Actin microfilaments adhere to the bone surface, forming an “actin ring”. Thus, osteoclasts are fastened to the bone surface with the formation of a sealing zone, separating resorption territory from extracellular fluid [75,76]. Within the territory delineated by the actin ring, the plasma membrane forms a highly convoluted ruffled border [77]. This is subdivided into a peripheral fusion and inner uptake zones. Binding of the endo/lysosomal membranes with the plasma membrane occurs in the fusion zone, while in the inner uptake zone, internalization of the degraded bone matrix takes place. Osteoclasts dissolve bone hydoxyapatite due to the secretion of protons through vacuolar adenosine triphosphatase (vATPase) into the peripheral fusion zone. In this manner, an acidified extracellular compartment emerges, in which lysosomal enzymes are secreted. Thus, the fusion zone becomes similar to the secondary lysosome, with acid hydrolases and a low pH needed for their activity, [78,79]. The pH in the erosion sites formed by osteoclasts may be as low as 4.7 [80]. Organic material is digested by cysteine proteinases and matrix metalloproteinases [25,81]. Osteoclasts also contain matriptase, a member of the type-2 transmembrane serine protease (TTSP) family [82], activated at low pH [83].

The release of growth factors presumably occurs during the digestion of decalcified bone matrix osteoclasts. Friess et al. [84] found that recombinant human BMP2 (rhBMP2) is well soluble at a pH of 4.5; conceivably, native BMPs released during matrix digestion dissolve in the acidified fluid present in the sealing zone. Products arising after matrix degradation are transcytosed and released at the secretory domain located at the opposite pole of the osteoclast into the extracellular matrix. Then, some of them may pass into blood vessels [3,85], while others, i.e., growth factors, presumably act on the local osteoprogenitor cells and stimulate bone deposition on the remains of the calcified longitudinal septa (Figure 3). Endocytosis and transcytosis from the ruffled border occur rapidly and the half-life of the endocytosed material inside the cells lasts 22 min [14].

## 10. Protection of Growth Factors against Denaturation by Proteolytic Enzymes

The question remains as to how growth factors are protected against proteolytic enzymes acting within the sealing zone. All BMPs contain a cystine knot involving six cysteine residues and form homodimers where two BMP molecules are joined by a disulphide bridge through a seventh cysteine residue [86]. A similar cystine-knot is present in miniproteins and is considered to be responsible for their exceptional stability towards thermic or proteolytic degradation [87,88]. Thus, it is conceivable that the presence of the cystine knot protects BMPs during transport from the sealing zone to the extracellular space.

It may also be significant that BMPs and GDFs bind to heparin and heparin/heparan sulphate [89]. The heparan sulphate (HS) chain of HSPGs (heparan sulphate proteoglycans) is a linear polysaccharide that supplies docking sites for the HS-binding ligands via electrostatic interactions [90]. Thus, growth factors released by osteoclasts from the bone matrix could bind to heparan sulphate in the ruffled membrane of the inner uptake zone and be transported by endocytosis to the secretory domain. Interestingly, Kim et al. [91] found that upon incubation of rhBMP2 and rhBMP7 with Chinese hamster ovary (CHO) cells, both agents bonded to the cell surface HSPGs, but only rhBMP2 was actively internalized. Native BMPs may behave differently than recombinant forms, and osteoclasts obviously differ from CHO cells; nevertheless, the mechanism of growth factor transport by osteoclasts in view of the results of Kim et al. [91] may be more complex than suggested above.

## 11. Formation of Osteoclasts Resorbing Calcified Cartilage Is Stimulated by Factors Released from Non-Calcified Cartilage by Septoclasts

While osteoclasts may release and transport growth factors deposited in a calcified cartilage matrix, the stimulus for their formation may originate from the non-calcified matrix accompanying the former. Non-calcified cartilage in the zone of provisional calcification is resorbed by septoclasts, the cells accompanying the capillary sprouts growing from metaphysis and characterized by a high content of cathepsin B [92,93] and selective expression of epidermal-type fatty acid-binding protein (E-FABP/FABP5) [94]. Septoclasts also displayed the presence of PDGF receptor beta (PDGFRβ) [94] as well as PDGF-BB [95]. Septoclasts have a mesenchymal origin arising from pericytes accompanying capillary vessels and, contrary to osteoclasts, do not arise from haematopoietic cells [94,96]. Septoclast specification requires the presence of the Notch ligand Delta-like 4, produced by endothelial cells, [96].

Septoclasts develop a ruffled border extending into the non-calcified cartilage and degrading the last transverse septum of the rows of chondrocytes in the growth plate [92,93]. They express lysosomal-associated membrane protein 1 (LAMP1) and proteolytic enzymes MMP9 and MMP14 [96]. Thus, it seems possible that septoclasts can liberate growth factors from the non-calcified matrix. The question remains as to how they transport them to the responding cells: could it occur in a manner similar to that in osteoclasts?

A panel of growth factors were released from the calf costochondral junction during hyaluronidase digestion [17], thus originating from the non-calcified cartilage, predominated by NELL1; BMP7 and GDF5, in addition to VEGF, were also present in a noticeable quantity. VEGF not only provokes the formation of blood vessels required for epiphyseal cartilage nutrition [97] but also stimulates the differentiation of monocytes to osteoclasts [98]. Arising osteoclasts would digest the calcified matrix and release growth factors stimulating bone formation (Figure 3).

## 12. Isolation of Growth Factors from Calf Costochondral Junctions

The convenient material for the study of growth factors may serve rib costochondral junctions, owing to their similarity in structure and function in the epiphyseal cartilage of long bones [68,99]. Particularly useful are calf ribs in view of their size and availability of bovine ELISA (enzyme-linked immunosorbent assay) tests for numerous bovine growth factors [17]. The evaluation of the profile of the growth factors, which accumulated in a calcified and adhering non-calcified matrix from calf costochondral junctions, demonstrated the quantitative predominance of NELL1, BMP7 and GDF5 (also known as BMP14, [53]), as well as the presence of BMP2, BMP3, BMP4, TGFβ1 and bFGF [17] (Figure 3).

## 13. Growth Factors Responsible for Initial Bone Formation in Human Metaphysis Remain *t**erra incognita*—Could Analysis of BMPs mRNA Expression in Chondrocytes from Human Epiphyseal Cartilage Help to Identify Them?

Our observations on the growth factors’ quantitative content in calf costochondral junctions indicate that, at least in this species, the differentiation of osteoprogenitor cells into osteoblasts and subsequent bone deposition may be under the control of at least three factors, i.e., NELL1 BMP7 and GDF5, with the possible contribution of other BMPs [17]. Unfortunately, data on the presence of the growth factors at the site of initial bone formation in the metaphysis of other species are lacking. They could be obtained for other large animals but in the case of small ones, collecting the sufficient material for analysis could be difficult. We have tried to overcome this obstacle through the analysis of BMPs mRNA expression in chondrocytes from the epiphyseal cartilage of the costochondral junction, assuming that the level of expression would be proportional to the amount of deposited growth factor. Chondrocytes isolated from proliferation, the hypertrophic zones of young rats, predominantly expressed BMP2, BMP5, BMP6, and BMP7 [16]. Nillson et al. [100] microdissected chondrocytes from the resting, proliferative, proliferative-hypertrophic transition, and hypertrophic zone of proximal tibial epiphyses of 7-day-old rats and found that mRNAs of BMP2, BMP6 and BMP7 predominated in all zones (BMP5 was not analysed). Thus, in rats, initial bone formation in metaphysis seems to be under the control of BMP2, BMP5, BMP6, and BMP7. In calves, it was predominated by NELL1, BMP7, and GDF5, while in rats, the last two factors were barely expressed [16,17].

Bal et al. [21] listed 13 papers analysing the influence of BMP2 associated with different carriers on the stimulation of bone formation in various animal models on the assumption that BMP2 may be optimal for human use. Our experiments suggest [16] that this factor could be the preferred choice for rat, but definitely not for bovine osteogenesis stimulation, since there seem to be distinct species differences in the panel of growth factors involved in physiological ossification (Figure 4).

Using BMPs for osteogenic stimulation, it is necessary to realize that various BMPs preferentially bind to different transmembrane receptors and that particular BMPs may not optimally fit to the cells to be stimulated. It also appears possible that the level of expression of various miRNAs varies depending on the type of BMPs. Thus, as long as the identity of human growth factors dominating during bone formation is not determined, studies with randomly chosen BMPs and animal osteoprogenitor cells may not be the most informative.

The question arises as to how much human material would be needed for the analysis of mRNA growth factors expression. Collagenase digestion of cartilage collected from the proliferative-hypertrophic zone of all costochondral junctions from two 6-week-old rats yielded about 2 × 10^6^ chondrocytes [16]. Taking into consideration the differences in size, presumably only a few human ribs would be sufficient for analysis.

## 14. Conclusions

Calcified cartilage present in the endochondral growth plate serves as the depot for growth factors produced by chondrocytes from the proliferative and hypertrophic zone and are responsible for the initiation of physiological bone formation. These growth factors seem to be species specific. Establishing the identity of the human growth factors directing ossification at the endochondral stage could favourably influence their choice for clinical practice.

Accumulating the large amount of human calcified epiphyseal cartilage needed for the quantitative study of numerous growth factors would be difficult. Collection of human chondrocytes from chondrocostal junctions for the study of growth factors expression seems, however, feasible and compatible with ethical standards. According to the Academy of Medical Royal Colleges, 2015, Recommendation 9: “When parents would like to donate their child’s organs for transplantation, but this is not clinically possible, clinicians should attempt wherever possible to accept such organs for research, if this is an acceptable alternative to the parents” [101].

The authors of this review do not have the facilities to collect tissues from human donors to study the growth factor production of human chondrocytes, but hope that the presentation of arguments in favour of such a study may attract attention and encourage competent laboratories to undertake the task.

## Figures and Tables

**Figure 1 ijms-23-10327-f001:**
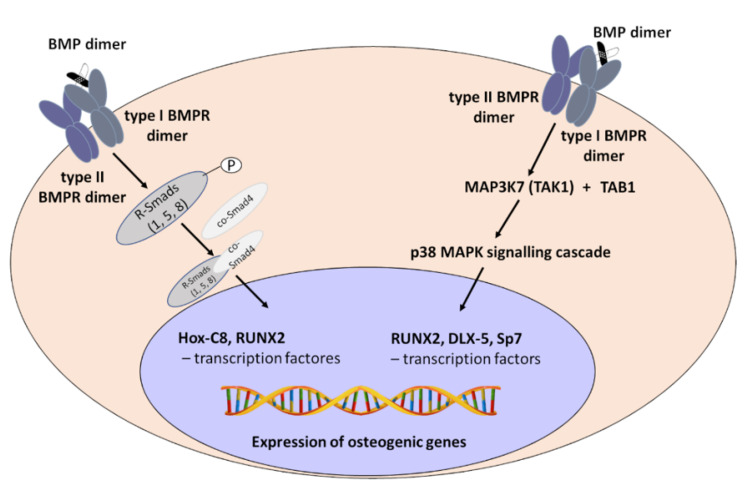
Canonical and non-canonical BMP signalling pathway.

**Figure 2 ijms-23-10327-f002:**
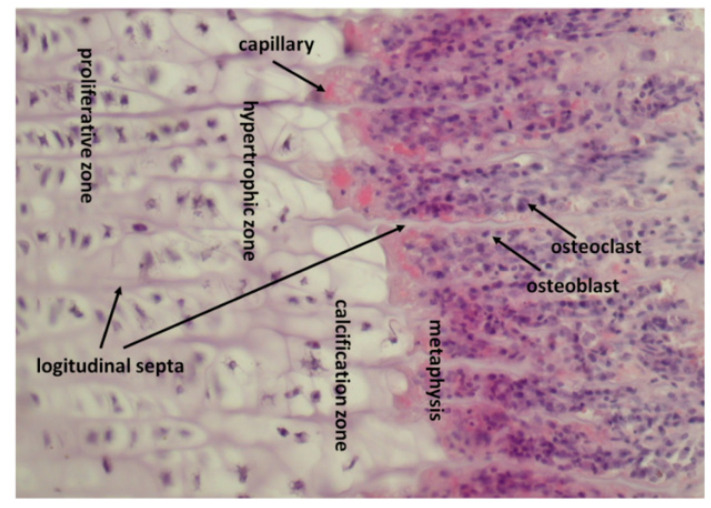
Structure of calf costochondral junction. Proliferative, hypertrophic and provisional calcification zones are present. The latter is connected with the metaphysis. Chondrocytes in the proliferative and hypertrophic zones form elongated columns separated by longitudinal septa formed by cartilage matrix. Chondrocytes within the columns are separated by transverse septa. In the zone of provisional calcification, longitudinal septa are calcified. Septoclasts occur around capillaries at the boundary of metaphysis and provisional calcification zone, but it is impossible to recognize them in haematoxylin and eosin staining. Septoclasts resorb the noncalcified transverse septa, whereas the calcified longitudinal septa are resorbed by osteoclasts. Osteoblasts deposit bone on the remnants of longitudinal septa. HE, magnification, ×200.

**Figure 3 ijms-23-10327-f003:**
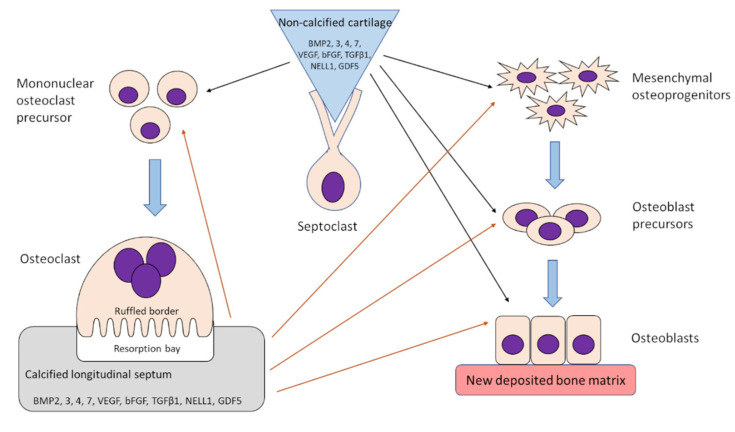
Tentative presentation of processes occurring during endochondral bone formation in calves. Septoclasts release growth factors from the non-calcified (black arrows) and osteoclasts from the calcified (red arrows) part of the longitudinal septa. These growth factors stimulate formation of osteoclasts from mononuclear precursors as well as differentiation of mesenchymal cells towards osteoprogenitors, osteoblast precursors and finally osteoblasts. The cooperation of septoclasts and osteoclasts during bone formation is probably similar also in other species, but the profile of growth factors present in non-calcified and calcified cartilage was, as yet, identified only in calves.

**Figure 4 ijms-23-10327-f004:**
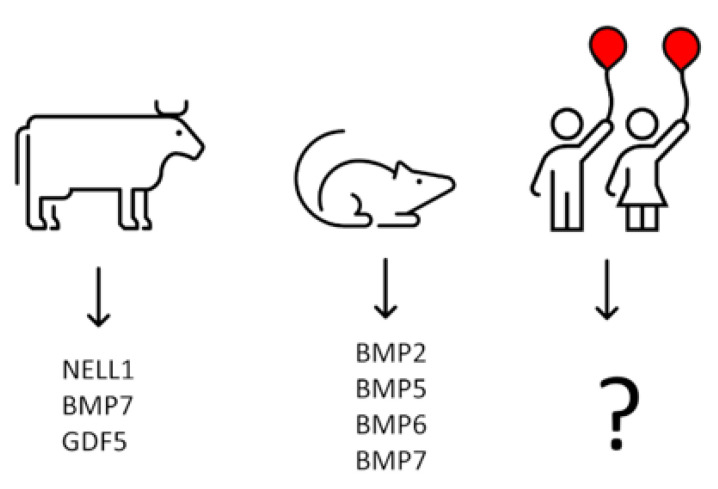
Profile of growth factors acting during physiological bone formation seems to be species specific.

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
