# Peer review of "Could BMPs Therapy Be Improved if BMPs Were Used in Composition Acting during Bone Formation in Endochondral Ossification?"

_ijms, 2022, doi:10.3390/ijms231810327_

Round 1

Reviewer 1 Report

The article presents a good idea.  Although the initial question is interesting, I have a few issues with the study.

First: There are still some mistakes in grammar and misprints, the authors should carefully check this manuscript. Please provide

This article fits into the framework of translational medicine: how to fill the gap between basic sciences and clinical sciences. How to understand  bone disorders based on bone morphogenetic proteins. The author can add a paragraph in the introduction to explain the idea in the context of translational medicine. Example of an article should be cited:

doi: 10.1080/03007995.2017.1385450

doi: 10.1097/BCO.0000000000000846

1. Title: good

2. Abstract: it captures the appropriate essence of the manuscript. Excellent.

3. Introduction: The introduction doesn't identify the problem that is being addressed in the manuscript and doesn't develop and states the purpose of the manuscript. .Please overview your (proposed) approach more accurately. Please also provide a rationale behind using the proposed approach, i.e., what are its benefits in comparison to alternative approaches. What are the main limitations of alternative approaches in comparison to the proposed approach? What is the gap that you aim to address with your approach?

4. Tables and figures: Quality of figures is so important too. Please provide some high-resolution figures. Some figures have a poor resolution.

5. References: I have verified all references and all key references are correct.

6. Discussion:

* The authors don't discuss the  limitations of the study.

7. Conclusion: The conclusion is not for asking questions. Please reformulate it and resume this manuscript and add future works.

8. I would like to know the point of view of the authors about the validation of this study and determine the accuracy of this technique in a clinical setting.

Author Response

Thanks for the review and valuable comments. We did our best to correct the manuscript according to the suggestions.

Reviewer writes:

“This article fits into the framework of translational medicine: how to fill the gap between basic sciences and clinical sciences. How to understand bone disorders based on bone morphogenetic proteins. The author can add a paragraph in the introduction to explain the idea in the context of translational medicine. Example of an article should be cited:

doi: 10.1080/03007995.2017.1385450

doi: 10.1097/BCO.0000000000000846

  1. Introduction: “The introduction doesn’t identify the problem that is being addressed in the manuscript and doesn’t develop and states the purpose of the manuscript. .Please overview your (proposed) approach more accurately. Please also provide a rationale behind using the proposed approach, i.e., what are its benefits in comparison to alternative approaches. What are the main limitations of alternative approaches in comparison to the proposed approach? What is the gap that you aim to address with your approach?”

We have changed Introduction and added a few sentences on translational medicine:

Introduction

Isolation of bone morphogenetic proteins from bovine material and production of their human equivalents by genetic engineering inspired hope that their application will considerably improve treatment of bone disorders  [1] BMPs and other growth factors act on mesenchymal stem cells (MSCs), stimulate their differentiation into osteoprogenitor cells as well as the differentiation of the latter into osteoblasts and osteoclasts [2-14] , thus they were obvious candidates  for clinical use. Unfortunately their application in numerous instances was disappointing and caused various harmful side effects [15] Our previous studies suggested that the panel of growth factors acting during initial stages of physiological bone formation is species specific [16,17]. Clinically used BMP-2 and 7, as well as several other growth factors, effectively produce bone in rats [1] but they choice for use in clinics was arbitrary since there were no data to decide which would be optimal for humans. During endochondral bone formation growth factors are produced by chondrocytes, deposited in cartilage calcified and non-calcified matrix, released, and serve for stimulation of osteogenesis. Determination which growth factors are produced by human chondrocytes and dominate during endochondral ossification could at least offer a hint which of them would be best for healing of bone disorders. In this review we outline processes occurring during initial stages of bone formation in hope that it will  encourage studies on the human chondrocytes obtained from young transplant donors. Describing mechanism of BMPs action, the role of BMPs in osteogenesis and osteoblast differentiation, we tried to further underline the need for recognition of factors specific for bone formation in humans. The use of such factors – if they were recognized – could improve the treatment of bone disorders being at present, as evaluated by specialists and described in the paragraph “BMPs in clinics - pros and cons” far from optimal.

Detection of growth factors responsible for physiological bone formation in humans could be realized in the framework of translational medicine. This procedure consists of three components: the scientists studying particular problems, the clinicians introducing results of basic studies into medical practice and community supplying funds for the realization of the programme [18,19] In this particular case, since material for growth factors detection would have to originate from human donors, the paramount importance would be the willingness of donor family to donate the tissues for medical studies.

  1. Tables and figures: “Quality of figures is so important too. Please provide some high-resolution figures. Some figures have a poor resolution”.

Figures are prepared in higher resolution (1000dpi)

  1. Discussion:

“The authors don’t discuss the limitations of the study”.

As an explanation this fragment is added to the Paragraph 13.

The question arises how much of human material would be needed for analysis of mRNA growth factors expression. Collagenase digestion of cartilage collected from proliferative-hypertrophic zone of all costochondral junctions from two 6 week-old rats yielded about 2x106 chondrocytes [16]. Taking into consideration the difference in size presumably only a few human ribs would be sufficient for analysis.

  1. Conclusion: “The conclusion is not for asking questions. Please reformulate it and resume this manuscript and add future works”.

It is modified as follows:

Calcified cartilage present in the endochondral growth plate serves as the depot for growth factors produced by chondrocytes from proliferative and hypertrophic zone and responsible for initiation of physiological bone formation. These growth factors seem to be species specific.

Establishment of identity of the human growth factors directing ossification at the endochondral stage could favorably influence their choice for clinical practice.

The rest of conclusions was not changed.

  1. “I would like to know the point of view of the authors about the validation of this study and determine the accuracy of this technique in a clinical setting”.

Determination of growth factor mRNA from isolated chondrocytes seems to be sufficiently precise to accept the obtained results as valid. Results obtained in our study with collagenase liberated chondrocytes are in good accord with results of Nillson et al., (quoted  as [100] in the Discussion paragraph 13 in revised manuscript) who dissected by micromanipulation chondrocytes from various zones of rat epiphyseal cartilage. Clinical validation of the study –establishing whether the use of growth factors discovered by analysis of mRNA is better than just application of BMP-2 or 7 would be difficult due to the number of observation needed for critical results. It is also possible that the study would find out that at least one of these factors dominate in humans. We now study effects of rat specific BMPs on rat osteoblasts isolated from calvariae to see whether there is synergy between various BMPs. There are lines of human transformed osteoblasts which respond to BMPs in vitro – they conceivably could also serve for similar experiments. Anyhow, the study of growth factors expressed in endochondral cartilage would increase basic knowledge of human physiology. Besides, it is astonishing how many studied on the effect of BMPs connected to various carriers are done in rabbits, dogs, sheep or monkeys without considering species specificity. Maybe this review would limit such carefree activity.

  1. Wozney, J.M. Overview of bone morphogenetic proteins. Spine 2002, 27, S2-8, doi:10.1097/00007632-200208151-00002.
  2. Amarasekara, D.S.; Kim, S.; Rho, J. Regulation of Osteoblast Differentiation by Cytokine Networks. International journal of molecular sciences 2021, 22, doi:10.3390/ijms22062851.
  3. Cappariello, A.; Maurizi, A.; Veeriah, V.; Teti, A. The Great Beauty of the osteoclast. Archives of biochemistry and biophysics 2014, 558, 70-78, doi:10.1016/j.abb.2014.06.017.
  4. Chan, W.C.W.; Tan, Z.; To, M.K.T.; Chan, D. Regulation and Role of Transcription Factors in Osteogenesis. International journal of molecular sciences 2021, 22, doi:10.3390/ijms22115445.
  5. Chen, G.; Deng, C.; Li, Y.P. TGF-β and BMP signaling in osteoblast differentiation and bone formation. International journal of biological sciences 2012, 8, 272-288, doi:10.7150/ijbs.2929.
  6. Cheng, H.; Jiang, W.; Phillips, F.M.; Haydon, R.C.; Peng, Y.; Zhou, L.; Luu, H.H.; An, N.; Breyer, B.; Vanichakarn, P.; et al. Osteogenic activity of the fourteen types of human bone morphogenetic proteins (BMPs). The Journal of bone and joint surgery. American volume 2003, 85, 1544-1552, doi:10.2106/00004623-200308000-00017.
  7. da Silva Madaleno, C.; Jatzlau, J.; Knaus, P. BMP signalling in a mechanical context - Implications for bone biology. Bone 2020, 137, 115416, doi:10.1016/j.bone.2020.115416.
  8. Halloran, D.; Durbano, H.W.; Nohe, A. Bone Morphogenetic Protein-2 in Development and Bone Homeostasis. Journal of developmental biology 2020, 8, doi:10.3390/jdb8030019.
  9. Koosha, E.; Eames, B.F. Two Modulators of Skeletal Development: BMPs and Proteoglycans. Journal of developmental biology 2022, 10, doi:10.3390/jdb10020015.
  10. Nickel, J.; Mueller, T.D. Specification of BMP Signaling. Cells 2019, 8, doi:10.3390/cells8121579.
  11. Ponzetti, M.; Rucci, N. Osteoblast Differentiation and Signaling: Established Concepts and Emerging Topics. International journal of molecular sciences 2021, 22, doi:10.3390/ijms22136651.
  12. Sampath, T.K.; Reddi, A.H. Discovery of bone morphogenetic proteins - A historical perspective. Bone 2020, 140, 115548, doi:10.1016/j.bone.2020.115548.
  13. Samsonraj, R.M.; Raghunath, M.; Nurcombe, V.; Hui, J.H.; van Wijnen, A.J.; Cool, S.M. Concise Review: Multifaceted Characterization of Human Mesenchymal Stem Cells for Use in Regenerative Medicine. Stem cells translational medicine 2017, 6, 2173-2185, doi:10.1002/sctm.17-0129.
  14. Stenbeck, G.; Horton, M.A. Endocytic trafficking in actively resorbing osteoclasts. Journal of cell science 2004, 117, 827-836, doi:10.1242/jcs.00935.
  15. James, A.W.; LaChaud, G.; Shen, J.; Asatrian, G.; Nguyen, V.; Zhang, X.; Ting, K.; Soo, C. A Review of the Clinical Side Effects of Bone Morphogenetic Protein-2. Tissue engineering. Part B, Reviews 2016, 22, 284-297, doi:10.1089/ten.TEB.2015.0357.
  16. Hyc, A.; Moskalewski, S.; Osiecka-Iwan, A. Growth factors in the initial stage of bone formation, analysis of their expression in chondrocytes from epiphyseal cartilage of rat costochondral junction. Folia histochemica et cytobiologica 2021, 59, 178-186, doi:10.5603/FHC.a2021.0017.
  17. Iwan, A.; Moskalewski, S.; Hyc, A. Growth factor profile in calcified cartilage from the metaphysis of a calf costochondral junction, the site of initial bone formation. Biomedical reports 2021, 14, 54, doi:10.3892/br.2021.1430.
  18. Mediouni, M.; D, R.S.; Madry, H.; Cucchiarini, M.; Rai, B. A review of translational medicine. The future paradigm: how can we connect the orthopedic dots better? Current medical research and opinion 2018, 34, 1217-1229, doi:10.1080/03007995.2017.1385450.
  19. Mediouni, M.; Madiouni, R.; Gardner, M.; N., V. Translational medicine: Challenges and new orthopaedic vision (Mediouni-Model). Curr Orthop Pract 2019, doi:10.1097/BCO.0000000000000846.

Reviewer 2 Report

This is a well written narrative review regarding the previous important findings on effects of growth factors, including the behavior of osteoclasts and osteoblasts.

This work covered the following topics and included a sufficient number of significant works:

Mechanism of BMP

BMPs in bone formation

Osteoblast differentiation

BMPs in clinical use

Synergistic effects of BMPs

Endochondral ossification

Roles of osteoclasts

Protection of growth factors

Formation of osteoclasts

and other few sections for specific roles.

This article did not have methods section and authors just introduced findings that they thought were important; however, the data they picked up encompassed majority of key findings in this field.

I am not sure if authors should put the following statement in Abstract:

Authors of this review do not have facilities to conduct such a study and can only appeal to competent institutions to undertake the task.

This statement could be shown only in the main text or in Conclusion section.

Author Response

Thanks for the review and valuable comments.

Reviewer writes: I am not sure if authors should put the following statement in Abstract: “Authors of this review do not have facilities to conduct such a study and can only appeal to competent institutions to undertake the task. This statement could be shown only in the main text or in conclusion section”.

On the other hand, the first reviewer considers the abstract as “excellent”. Thus we rather prefer to leave the abstract intact, not to risk the decrease of its “excellency”.

Round 2

Reviewer 1 Report

Accept